# Design, Fabrication, and Characterization of a Laser-Controlled Explosion-Initiating Device with Integrated Safe-and-Arm, EMP-Resistant, and Fast-Acting Technology Based on Photovoltaic Power Converter

**DOI:** 10.3390/mi13050728

**Published:** 2022-04-30

**Authors:** Guofu Yin, Huiqin Bao, Yulong Zhao, Wei Ren, Xiangfei Ji, Jianhua Cheng, Xi Ren

**Affiliations:** 1The School of Microelectronics, Xidian University, Xi’an 710071, China; 2Science and Technology on Applied Physical Chemistry Laboratory, Shanxi Applied Physic-Chemistry Research Institute, Xi’an 710061, China; rw0192@163.com (W.R.); jixiangfei213@yeah.net (X.J.); yinxiaopang2016@yeah.net (J.C.); renxi213@yeah.net (X.R.); 3Xi’an Mingde Institute of Technology School of Information Engineering, Xi’an 710124, China; 4State Key Laboratory for Manufacturing System Engineering, School of Mechanical Engineering, Xi’an Jiaotong University, Xi’an 710049, China; zhaoyulong@mail.xjtu.edu.cn

**Keywords:** intelligent initiation system, explosion initiating device, safe-and-arm, laser power converter, semiconductor bridge

## Abstract

To augment the intelligence and safety of a rocket or ammunition engine start, an intelligent initiation system needs to be included in the data link. A laser-controlled intelligent initiation system with inherent safety and a laser-controlled explosion-initiating device (LCEID) incorporating electromagnetic pulse (EMP) resistant, safe-and-arms fast-acting modular device based on photovoltaic power converter technology is designed and fabricated in this work. LCEID is an integrated multi-function module consisting of the optical beam expander, GaAs photovoltaic (PV) array, safe-and-arms integrated circuit, and low-energy initiator. These components contribute to EMP resistance, fast-acting, safe-and-arm, and reliable firing, respectively. To achieve intelligent initiation, each LCEID has a unique “identification information” and a “broadcast address” embedded in integrated-circuit read-only memory (ROM), which is controlled by encoded laser addressing. The GaAs PV array was investigated to meet the low-energy initiator firing voltage requirements. Experimental results show that the open-circuit voltage, short-circuit current, and maximum power output of the four-junction GaAs PV array illuminated by a 5.5 W/cm^2^ laser beam were 220 mA, 21.5 V, and 3.70 W, respectively. When the voltage of the 22 μF energy storage capacitor exceeds 20 V, the laser charging time is found to be shorter than 2.5 s. Other aspects of LCEID, such as laser energy coupling efficiency, the firing process, and the energy-boosting mechanism, were explored. Measurements show that the coupling efficiency of the micro lens with a radius of curvature D = 20 μm and size of r = 50 μm reaches a maximum of 93.5%. Furthermore, for more than 18 V charge voltage, the LCEID is found to perform reliably. The fabricated LCEID demonstrated a high level of integration and intrinsic safety, as well as a finely tailored initiation performance that could be useful in military applications.

## 1. Introduction

The first component of a rocket or ammunition system is the initiation system. Its safety and dependability are critical to the rocket or ammunition system’s overall performance. Rocket or ammunition systems are evolving towards informatics, intelligence, and networking as a result of the rapid growth of information and networking technology [1,2]. This development trend necessitates the initiation system’s intelligence [3,4]. An intelligent initiation system requires improved data exchange and information cross-linking between the ammunition and the ammunition’s automated attacking capabilities and also the damaging effect [5]. Laser-controlled initiation system is a special type of intelligent initiation system. It possesses intrinsic EMP resistance, which considerably increases the ammunition’s safety. As a result, it has piqued the interest of national defense personnel. A photoelectric integrated initiator based on a semiconductor bridge (SCB) has been proposed in reference [6]. It combines the photoelectric converter or laser’s output energy with the semiconductor bridge chip’s low-energy initiation feature. An insensitive detonator/initiator that is not easily to stimulate has also been designed. Authors of [7] have reported the design of an optically controlled smart initiator consisting of a tube-shell, a fiber-optic window, a photovoltaic (PV) integrated circuit, and a capacitor with an explosion sequence and a cover sheet. When the capacitor is charged to 25 V, the storage capacity can reach 6.25 mJ, which is enough to detonate cyclotetramethylenetetranitramine (HMX) or Titanium sub-hydride/potassium perchlorate (THKP). Thiokol company has designed a miniaturized optical-control initiation system [8]. Each initiator contains a solid-state microcircuit with a photoelectric converter, decoding module, time-control module, and fire capacitor. The system can be used for multiple-event control in multi-stage systems such as launch vehicles. A photoelectric SAFE/ARM initiation system was developed by Alliant Techsystems Ins. [9], which contains an initiation-control module, dual-fiber communication line, and photoelectric initiator. With this system, a laser-coded firing control was realized. In this paper, we focus on the integrated design, manufacturing, and characterization of a laser-controlled intelligent initiating system and LCEID based on PV power converter technology that integrates safe-and-arms, EMP resistance, and fast-acting technologies.

## 2. Laser-Controlled Intelligent Initiation System

Laser-controlled intelligent initiation system consists of a multiple-point LCEID, an optical fiber bus controller (OFBC), and an optical fiber bus network, as shown in Figure 1. The OFBC has two output modes which are continuous laser and encoded laser. The continuous laser is fed into the LCEID through an optical fiber bus network, where it illuminates the GaAs PV array, converting laser light energy to electric energy and allowing the LCEID to function. The “identification information” to be used in directly controlled LCEID is contained in an encoded laser, which uses a 16-bit control signal. By transmitting these lasers’ encoded commands, the OFBC controls all terminal LCEID on the optical fiber bus according to the initiation criteria of the host controller. Only the LCEID‘s “identification information” corresponds to that which is stored in the integrated ROM and derived from the received command in strict accordance with the data interaction protocol. The built-in test and initiation functions run on their own automatically, following predetermined logic. The OFBC combines laser light energy and PV technology to safely and reliably initiate the LCEID. The center wavelength, maximum output power, numerical aperture and fiber core diameter of the laser embedded in the OPBC are 808 nm, 10 W, 0.22 NA, and 200 μm, respectively.

The SCB electrical initiator is completely isolated from sources of energy that could cause unintended activation with this technology. All power, command, and data signals are transmitted optically between the OFBC and the LCEID by laser diodes via fiber optic cables. The optical signals are then converted to electrical signals by PV converters for decoding and further action.

Thus, the system avoids transmission loss over long cable lengths, which is a problem for direct laser ordnance initiation systems, as well as the shielding and noise penalties that come with electrical communications. The OFBC contains system input/output, self-diagnostic features, an arming plug, and visual safe/arm indicators. The LCEID contains Safe-and-Arms functionalities and the initiator squib, which are activated by coded optical signals from the OFBC. When the system is turned on, the LCEID charges a capacitor on the spot, storing the firing energy at the time of initiation. The LCEID discharges the capacitor to the initiator squib, causing it to fire, in response to the FIRE order from the OFBC. The LCEID will rapidly discharge the capacitor through bleed resistors in response to the SAFE command or the loss of signal from the SAFE, making the system SAFE.

This system has the following advantages: each function has its own safe and arm; the system is inexpensive; reduced weight and envelope requirements are realized; the system provides a single upgradable initiation control module to serve multiple functions; functioning time is quick; improved launch response is realized through BIT capability; it complies with MIL-STD-1901 standards for in-line ordnance; optical isolation protects against ESD, RFI, and EMP. The fiber optic cable eliminates explosive transfer assemblies, which provides safer handling, reduces explosive aging (service life) concerns, and improves routing flexibility between stages, as well as coding and multiple inhibits for greater safety.

## 3. Design of Laser-Controlled Explosion-Initiating Device

LCEID is the core component of the laser controlled intelligent initiation system. As shown in Figure 2a, it consists of an optical beam expander, GaAs PV array, storage circuit, information embedded microcontroller, fire circuit, SCB energy chip, and explosion sequence. To improve the safety and reliability, it is designed in modular form and requires an integrated fabrication, as shown in Figure 2b. The received continuous laser signal is converted to electricity within the LCEID and is used to charge the firing capacitor via the GaAs PV array. The firing capacitor has a built-in bleed for discharge in the event when the initiation is halted. The firing capacitor cannot discharge energy to the SCB initiator unless a properly coded 16-bit firing signal is received. Decoding signals, setting timing, status monitoring, and controlling capacitor discharge are all done with a solid-state microelectronic device. In the ROM of the information embedded micro-controller, each LCEID has a unique “identification information” and a “broadcast address”. Moreover, LCEID possesses the built-in-test and self-initiation functions by accessing laser addressable control.

### 3.1. Design and Characterization of the Optical Beam Expander

It is well-known that the traditional electro-explosive device, as shown in Figure 3, uses electrical pins as the coupling window of the initiation energy and control signal. The ESD, RFI, and EMP interference will be very easy to couple to the sensitive device SCB or bridge wire through the electrical pin which will possibly cause accidental initiation and greatly reduce its safety. Electrical pins are commonly used as the coupling window of the initiation energy and control signal in traditional electro-explosive devices, as shown in Figure 3. ESD, RFI, and EMP interference will be very easy to couple to the sensitive device SCB or bridge wire through the electrical pin, resulting in accidental initiation and a significant reduction in its safety.

Optical components such as a self-focusing lens and a micro convex lens array are employed in this study to create a coupling window for laser energy and control signals from the OFBC, as illustrated in Figure 4. It will perform important functions as follows.

(1) Because of its separation from the external environment ESD, RFI, and EMP interference for electromagnetic sensitive devices such as SCB, it will improve the safety of the LCEID when compared to standard electro-explosive devices.

(2) Expanding and collimating the laser beam to enhance the photoelectric conversion efficiency.

Because the numerical aperture of the optical fiber is limited, the laser beam produced by the OFBC is very concentrated. As a result, being sufficiently and consistently linked to the PV array is difficult. To improve the coupling efficiency of laser energy and photoelectric conversion efficiency, the laser beam must be enlarged and collimated to ensure the power density uniformity of the laser irradiated to each unit PV converter on the GaAs PV array.

We focus on the coupling efficiency of the optical beam expander that couples on the PV array converter. The effects of the micro lens’ radius of curvature and the lens’s size on the coupling efficiency of the optical beam expander were then investigated. Figure 5 shows the coupling efficiency of each field of view with the radius of curvature of the micro lens unit when the unit diameter is D = 20 μm. When the radius of curvature of the micro lens unit is in the range of 28–50 μm, the coupling efficiency of each field of view of the optical beam expander increases with the increase of the radius of curvature. When the radius of curvature is r < 28 μm, the coupling efficiency of each field of view is less than 50%; when the radius of the micro lens unit is set to r = 50 μm, the coupling efficiency of the coupling mechanism reaches its height.

Figure 6 shows the coupling efficiency of the three fields of view changes with different cell sizes when the radius r = 50 μm. Starting from the minimum size D = 20 μm, as the size of the micro lens unit increases, the coupling efficiency of each field of view of the optical beam expander shows a downward trend. When D = 35 μm, the coupling efficiency of each field of view is close to 50%.

Comprehensive analysis shows that the smaller the size of the micro lens array unit and the larger the radius of curvature, the higher the coupling efficiency of the coupling mechanism. When D = 20 μm and r = 50 μm, the coupling efficiency of the optical beam expander reaches the maximum η = 93.5%. This will greatly improve the energy utilization of the LCEID.

### 3.2. Design and Fabrication of GaAs PV Converter

GaAs PV array is the most critical component of LCEID. Its performance has a direct impact on LCEID’s working power and initiation performance. Therefore, we focus on the design, fabrication, and performance of the GaAs PV array. MOCVD (Metal Organic Chemical Vapor Deposition) [10,11] is used to fabricate the GaAs PV converter, which has dimensions of 10 mm × 10 mm. To produce the desired output voltage, individual converters are frequently joined in a series string of converters.

The scheme of the optimum structure of GaAs PV converters grown is shown in Figure 7, which consists of n-GaAs substrate (5 × 10^18^ cm^−3^, 350 μm), n-GaAs buffer (5 × 10^18^ cm^−3^, 1.0 μm), n-AlGaAs BSF (Back Surface Field) (5 × 10^18^ cm^−3^, 0.05 μm), n-GaAs base (1 × 10^17^ cm^−3^, 3.5 μm), p-GaAs emitter (2 × 10^18^ cm^−3^, 0.5 μm), p-GaInP window (5 × 10^18^ cm^−3^, 0.05 μm), and p++-GaAs cap layer (5 × 10^19^ cm^−3^, 0.5 μm). The GaAs cap layer allows the top grid pattern to make excellent ohmic contact while minimizing the potential of grid metallization migrating into the semiconductor materials during processing [12,13,14]. The cap layer is selectively etched away between the grid lines before the application of the anti-reflection coating. The anti-reflection coating consists of a double layer of TiO_2_ and SiO_2_ optimized for minimum reflection under illumination with an 808 nm laser [15,16].

The PV converter is known to be nonlinear, with a single operating point where it produces the maximum power. The steady-state I-V characteristics of a p–n junction PV converter are often described based on one diode model [17,18]. Figure 8 depicts the theoretical circuit model of the PV converter.

Based on the p–n junction recombination mechanism, the output current of a PV converter can be expressed by Equation (1) that includes a light-generating current source, series resistance, and shunt resistance of diode with the diode equation shown by Equation (2). Where *q* is the electron charge, *K* is the Boltzmann^’^s constant, *T* is the temperature, $I_{o}$ is the reverse saturation current, n is the ideality factor of the diode, *I_ph_* is the light-generated photocurrent, *R*_s_ is the series resistance, and *R_sh_* is the shunt resistance.
(1)$$I=I_{ph}-I_{o}\left(\exp\frac{q\left(V+IR_{s}\right)}{nKT}-1\right)-\frac{V+IR_{s}}{R_{sh}}$$
(2)$$I_{d}=I_{o}\left(\exp\frac{qV_{d}}{nKT}-1\right)$$

Generally, the parallel resistance of a PV cell is large, so the third item of Equation (1) can be ignored, and can be simplified as:(3)$$I=I_{ph}-I_{o}\left(\exp\frac{q\left(V+IR_{s}\right)}{nKT}-1\right)$$

After transformation:(4)$$V=\frac{nkT}{q}\ln\left(\frac{I_{ph}-I}{I_{o}}+1\right)-IR_{s}$$

In this LCEID, the SCB initiator is an explosion-initiating device. The LCEID not only stores laser energy quickly but also ensures a consistent fire voltage for the SCB chip. Then, the output voltage *V_out_* of the GaAs PV converter must reach the minimum full fire voltage *V_min_* of the SCB initiator. A large number of experiments have shown that the minimum full fire voltage *V_min_* of the Si-based Al/CuO SCB is approximately 15 V [19,20]. As a result, a GaAs PV array with a serial-parallel connectivity topology was designed and built, which included a larger PV converter. Figure 9 shows that four-unit PV converters are connected in series and parallel from a monolithic block. In the ideal condition, the unit PV converter PV_1_~PV_4_ has equal photo-generated current, i.e., *I* = 2*I_ph_* = 2*I_ph1_* ~ = *2I_ph4_.*

Due to the same current through the series components PV_1_||PV_2__,_PV_3_||PV_4_, *I_ph_ =*
*I_ph1_* = *I_ph2_*, *I_ph_ =*
*I_ph3_* = *I_ph4_*. The series components (PV_1_||PV_2_) & (PV_3_||PV_4_) have equal output voltage, *V_out_ = V_1_* + *V_2_ = V_3_* + *V_4_*. This voltage is the GaAs PV array output voltage *V_out_*:(5)$$V_{out}=V_{1}+V_{2}=V_{3}+V_{4}=\frac{2nkT}{q}\ln\left(\frac{I_{ph}-I}{I_{o}}+1\right)-2IR_{s}$$
where q is the electron charge, *K* is the Boltzmann’s constant, *T* is the temperature, *I_o_* is the reverse saturation current, *n* is the ideality factor of the diode, *I_ph_* is the light-generated photocurrent, *R_s_* is the series resistance.

The output voltage and current of GaAs PV array are independent of the variation of *R_s_*. The variation of *R_s_* has a great influence on the curvature of the I-V curve near the maximum output power point of the GaAs PV array [12,22]. This means the smaller *R_s_* is, the greater the curve curvature is. That is to say, the output voltage and current of GaAs PV array with the same structure are related to the photoelectric current and temperature. They only depend on the laser power density which has been designed for the optical beam expander.

We designed and fabricated the GaAs PV array converter which is composed of four-junction sector structure units GaAs PV, a ceramic substrate, and gold-plated copper electrode, as shown in Figure 10. The ceramic substrate is used to dissipate the heat being generated during photoelectric conversion and improves the photoelectric conversion efficiency of the GaAs PV array.

The I-V and P-V characterization of the GaAs PV array converter was studied. The incident monochromatic light power density was fixed at 5.5 W/cm^2^. The open-circuit voltage of this structure was as high as 21.5 V, the corresponding short circuit current was about 220 mA, as shown in Figure 11. It can achieve the maximum output electrical power of 3.70 W. The final conversion efficiency was about 67%. This is a very high photoelectric current conversion efficiency compared to a conventional PV array. This structure provides higher performance features such as power production and conversion efficiency, as may be observed from the I-V curve.

### 3.3. Design of Safe-and-Arms Control Integrated Circuit

To improve the safety of LCEID, we designed the safe-and-arms control integrated circuit which was integrated into the bidirectional optical data converter, micro-control, Safe/Arm switch, firing capacitor, firing switch to improve the safety of LCEID, as shown in Figure 12a,b.

The LCEID has three modes of operation: SAFE, ARM, and FIRE. It can be commanded from safe to arm, arm to safe, and arm to fire. In the safe mode, the ARM/SAFE switch is in the safe state, which discharges or holds the discharged firing capacitor. In the arm mode, this switch is set to the armed state, which removes the discharge path and connects the power source to the firing capacitor, allowing it to charge. The microcontroller monitors the voltage on the firing capacitor and reports the charge voltage to the OFBC. This process makes the LCEID possess the built-in-test (BIT) capability. BIT provides a real-time system check and feedback the safe/arm status to the user both visually and through vehicle telemetry. Once the OFBC is in arm mode, it can be commanded to fire. The firing of the device takes place when the LCEID microcontroller closes the fire switches.

In the ROM of the information encoding microcontroller a unique “identification information” and “broadcast address” is embedded. The OFBC outputs the 16-bit bidirectional laser data which contains 2-bit “identity information” or “broadcast address” to address control LCEID and realizes the intelligent initiation.

## 4. Experiment Results and Discussions

### 4.1. Characterization of Laser Energy Storage

LCEID, being the weapon system’s first initiation component, must have a fast response feature. The laser’s energy storage performance thus plays a crucial role in its rapid response. Figure 13 and Figure 14 show the relationship between the firing capacitor voltage and the laser charging time under different laser power densities. If the charge time of the storage capacitor is required to be less than 3 s, the laser power density must be greater than 3.5 W/cm^2^. The higher the laser power, the faster the charging time. When the laser power density is 5 W/cm^2^, the charging time is 2.5 s. That means LCEID can store energy quickly, which is useful for improving the speed of an initiation system. However, with the increase of laser power, thermal damage to the GaAs PV array as reference [23]. As a result, the photoelectric conversion efficiency is dramatically reduced and the output voltage falls short of the integrated energy device’s performance requirements.

### 4.2. Characterization of Initiation

The main purpose of the LCEID-integrated multiple functional modules is to realize intelligent control and high-security features. However, the initiation effect of the SCB chip under the excitation of the firing capacitor discharge pulse energy is practically the same [24,25,26].

In this research, the firing capacitor energy is only affected by the laser power density without considering the influence of the GaAs PV array. Therefore, the initiation performance of LCEID is only affected by the laser power density. Figure 15 shows the schematic diagram of the LCEID initiation experiment. This experiment utilizes a laser with a power density of 5 W/cm^2^ to illuminate the GaAs PV array and decides the output energy. It is stored in the firing capacitor to provide energy for integrated energy device initiation. The PMOSFET is turned on under the control of the laser pulse signal with a specific frequency and forms the discharge circuit between the firing capacitor and SCB. The firing capacitor releases the electric energy to activate SCB initiation. Figure 16 shows the low-energy SCB experimental setup.

In this study, the LCEID was activated by using the energy stored in a 22 μF capacitor with a voltage of 20 V. The voltage and current measured across the LCEID lead during the test are depicted in Figure 17. The resistance and energy dissipated in the low-energy SCB were calculated from the measured voltage and current. Here, *t*_0_ was the starting time. Two small peaks in the first set of peaks of voltage were considered as *t*_11_ and *t*_12_. Owing to resistive heating, the temperature of the low energy SCB increases, and solid SCB melts and transformed into liquid. The maximum value of the second peak corresponded to *t*_3_. At the *t*_2_–*t*_3_ stage, the rapidly growing voltage initiates a discharge in gaseous SCB and plasma formation. *t*_2_ was the valley between the first and second peaks. At the *t*_1_–*t*_2_ stage, liquid silicon evaporates and is transformed into gaseous silicon. *t*_4_ was the moment that the current dropped to 0 A. We could relate features in plot to those of the photograph. From Figure 17, the five points were 0.92 μs, 2.05 μs, 3.48 μs, 4.00 μs*,* and 39.26 μs. In addition, the curve of energy dissipated and dynamic impedance is almost similar. Therefore, we considered the laser rapid energy storage would not change the firing process of SCB, which also experienced the processes of heating, melting, vaporization, and plasma generation.

As shown in Figure 18, the burst times under different excitation voltages were 3.42 s, 4.21 s, and 5.14 s. With increasing voltage, the time of the integrated energy device began to shorten, which is following the law of electron temperature and density variation. Figure 19 shows the result of the LCEID with different excitation voltages. With the rise in voltage, the integrated energy device’s bridge area is explosive enough. This means that as the voltage is increased, the reliability improves.

## 5. Conclusions

An LCEID with integrated safe-and-arm, EMP-resistant, and fast-acting tecnologies was designed and fabricated using modular design and integrated circuit fabrication techniques. By using the laser beam expanding and collimating method, the optical components are used to create a coupling window for the laser energy and a control signal that will isolate the external environment ESD, RFI, and EMP interference for electromagnetic sensitive SCB and increase coupling efficiency to a maximum of 93.5%. The four-junction GaAs PV array was designed and fabricated by using serial-parallel interconnection technology. The short-circuit current, open-circuit voltage, maximum power output, and conversion efficiency of the GaAs PV array when illuminated with a laser beam of 5.5 W/cm^2^ were 220 mA, 21.5 V, 3.70 W, and 67%, respectively. The safe-and-arm control integrated circuit has three modes of operation—SAFE, ARM, and FIRE—to ensure safety. It has been embedded with a unique “identity information” and a “broadcast address” which can be addressed and controlled by 16-bit bidirectional laser data to realize the intelligent initiation. Furthermore, a laser-controlled intelligent initiation system was designed based on the LCEID. In this system, the charging time is 2.5 s for the LCEID when the laser power density is 5 W/cm^2^ and the burst time under 22 V voltage excitation voltages was 3.42 s. This fast-acting system can potentially be used in civilian and military applications, especially in noncontact initiation such as airbags in a rocket engine, propulsion systems, and other ordnance systems.

## Figures and Tables

**Figure 1 micromachines-13-00728-f001:**
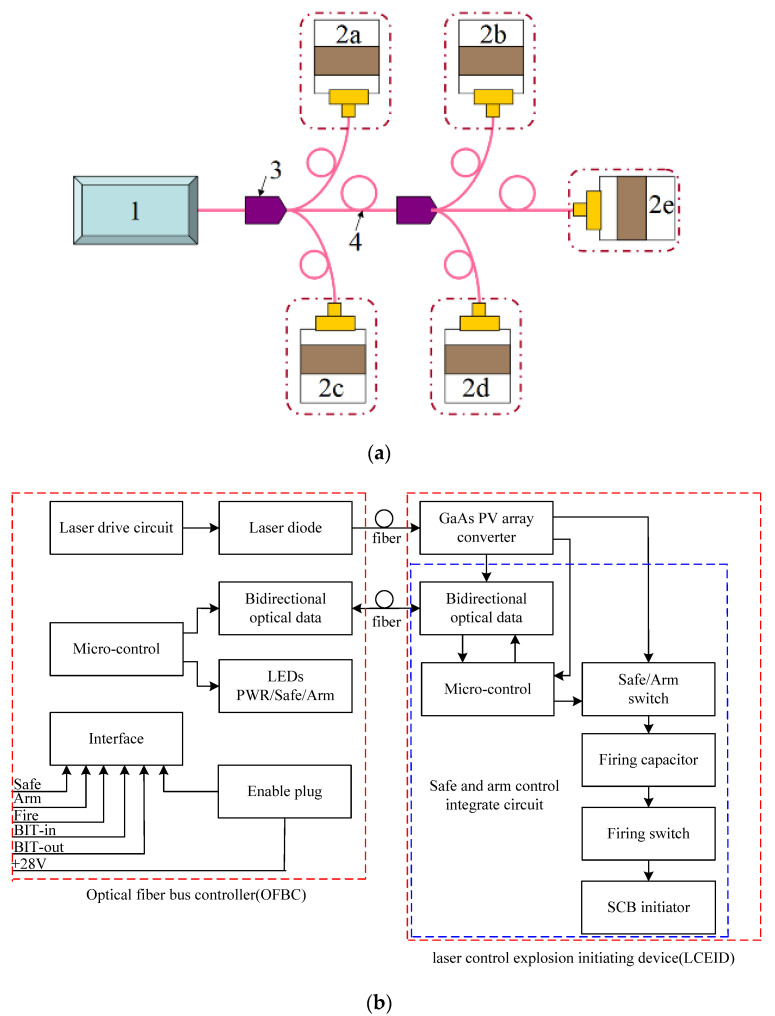
Schematic diagram of laser-controlled intelligent initiation system. (**a**) Multi-point initiation system: 1—OFBC; 2a~2e—LCEID; 3—optical coupler; 4—optical fiber bus network. (**b**) Schematic of single-point initiation system.

**Figure 2 micromachines-13-00728-f002:**
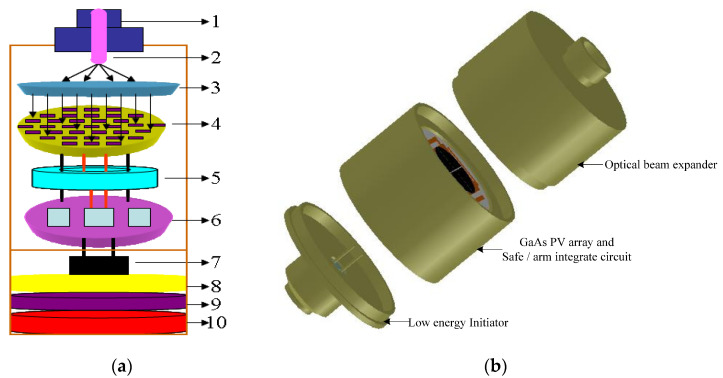
Schematic diagram of an LCEID. (**a**) Module composition.:1—FC/PC; 2—fiber; 3—optical beam expander; 4—GaAs PV array; 5—stored energy circuit; 6—safe-and-arms control integrated circuit; 7—SCB; 8—lead styphnate; 9—lead azide; 10—HMX. (**b**) Integrated structure.

**Figure 3 micromachines-13-00728-f003:**
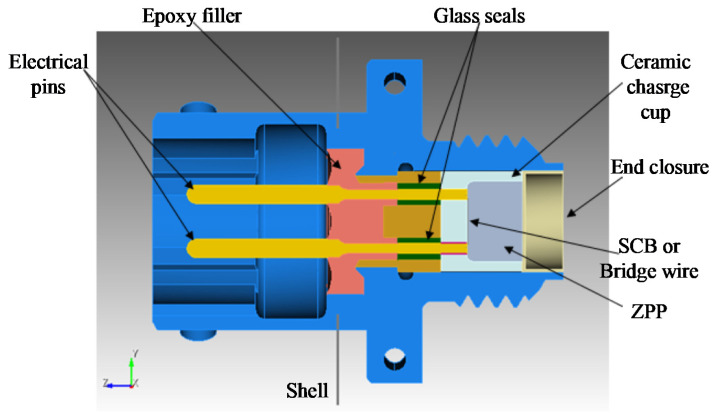
Traditional electro-explosive device.

**Figure 4 micromachines-13-00728-f004:**
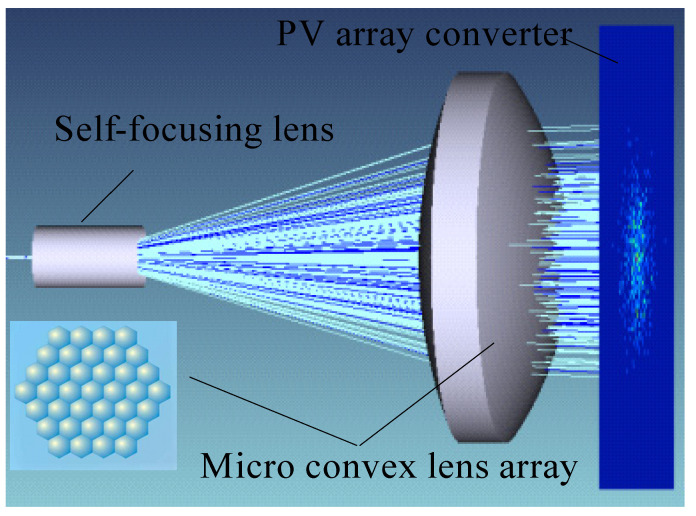
Optical beam expander.

**Figure 5 micromachines-13-00728-f005:**
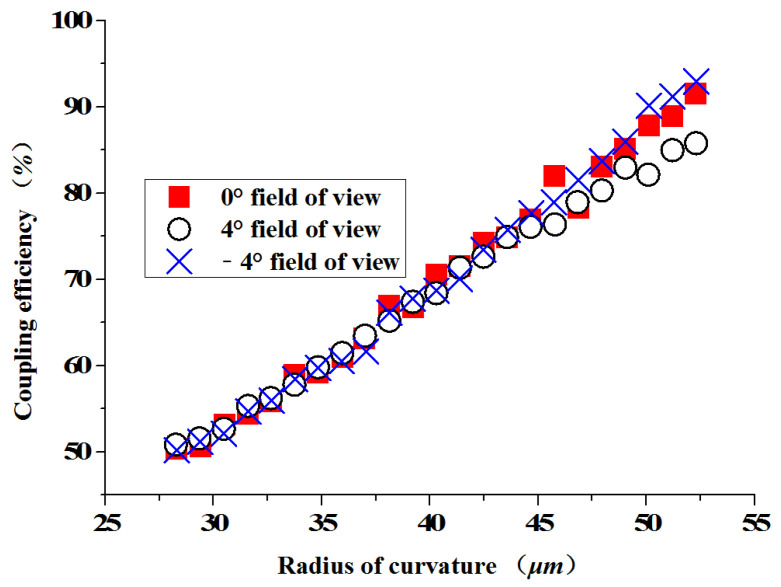
The radius of curvature of the micro lens and the coupling efficiency.

**Figure 6 micromachines-13-00728-f006:**
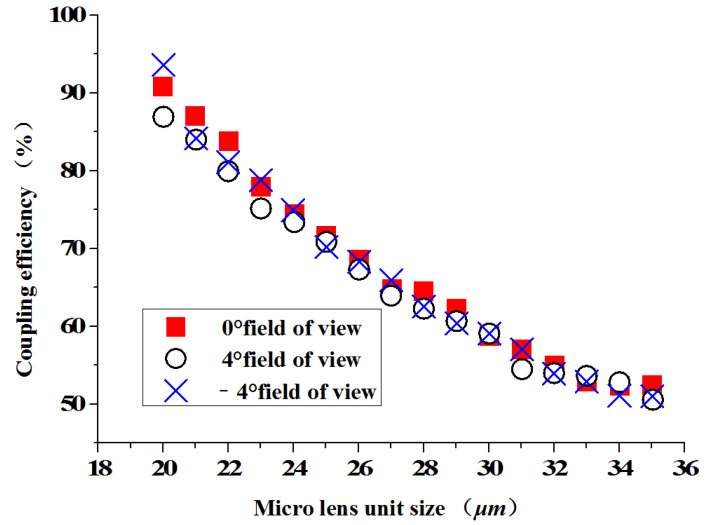
The size of the micro lens unit and the coupling efficiency.

**Figure 7 micromachines-13-00728-f007:**
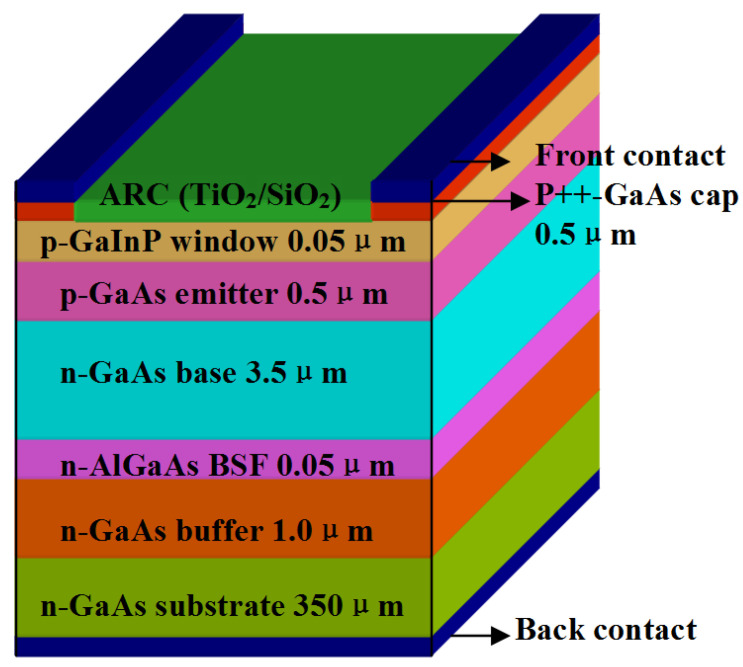
Structure of GaAs PV converter.

**Figure 8 micromachines-13-00728-f008:**
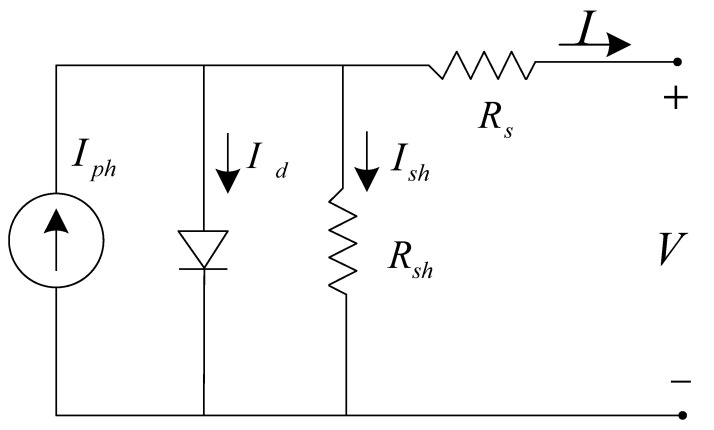
Theoretical circuit model of the PV converter.

**Figure 9 micromachines-13-00728-f009:**
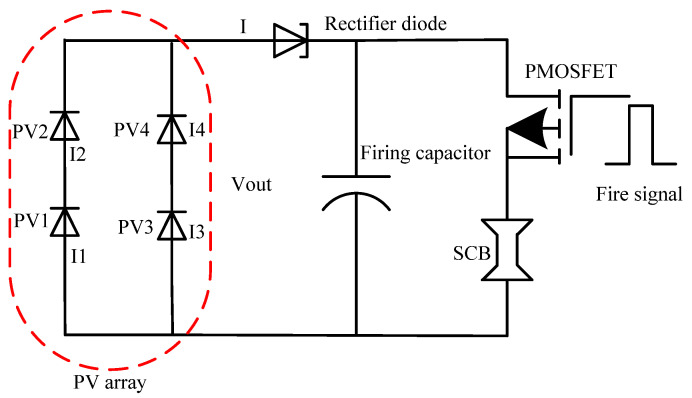
Laser storage and fire circuit of LCEID. Annotation: each PV consists of ten-junction vertically-stacked GaAs cells, as in reference [21].

**Figure 10 micromachines-13-00728-f010:**
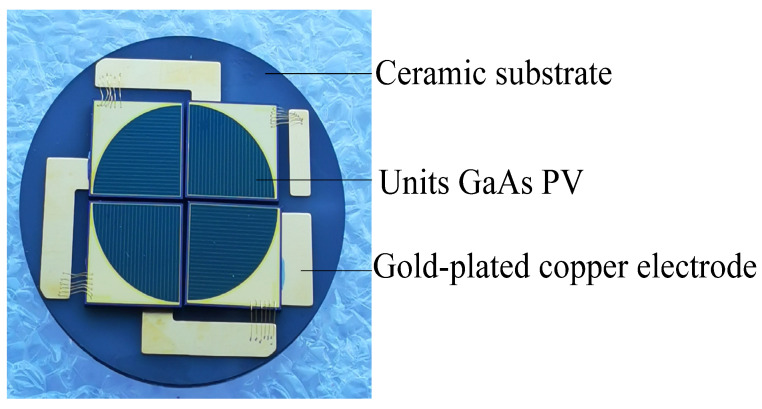
GaAs PV array converter.

**Figure 11 micromachines-13-00728-f011:**
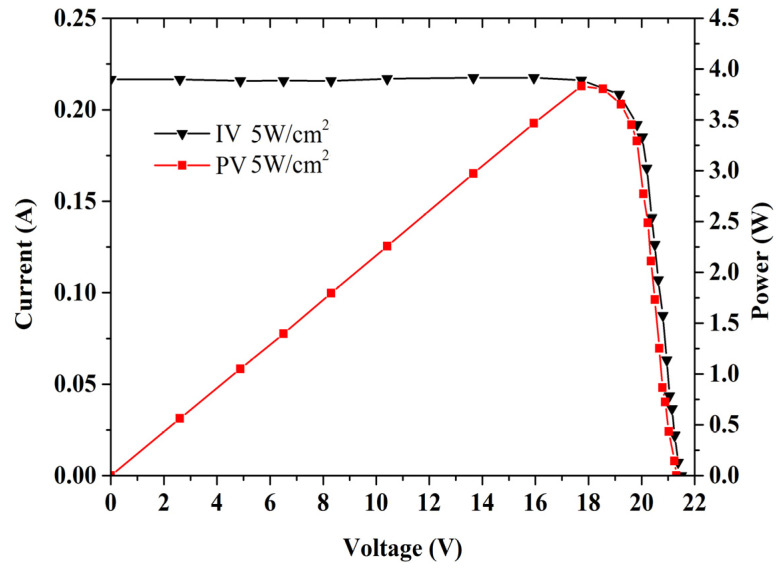
I-V and P-V curves of GaAs PV array converter.

**Figure 12 micromachines-13-00728-f012:**
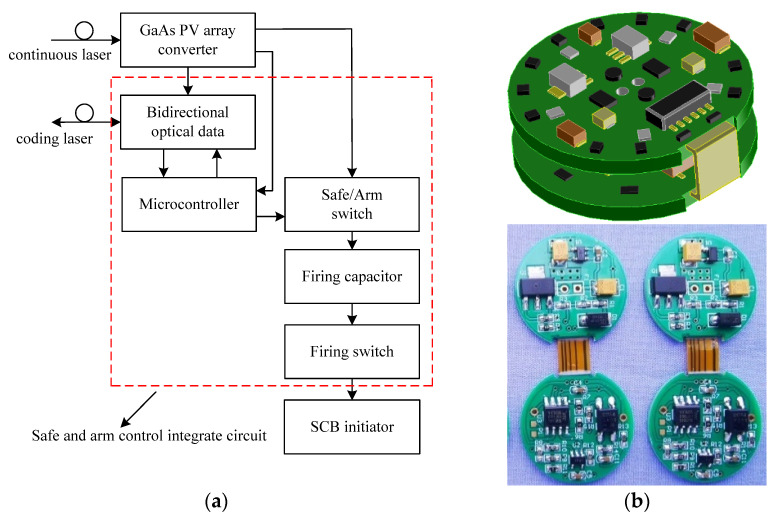
Safe-and-arm control-integrated circuit. (**a**) Module composition. (**b**) Integrated circuit prototype.

**Figure 13 micromachines-13-00728-f013:**
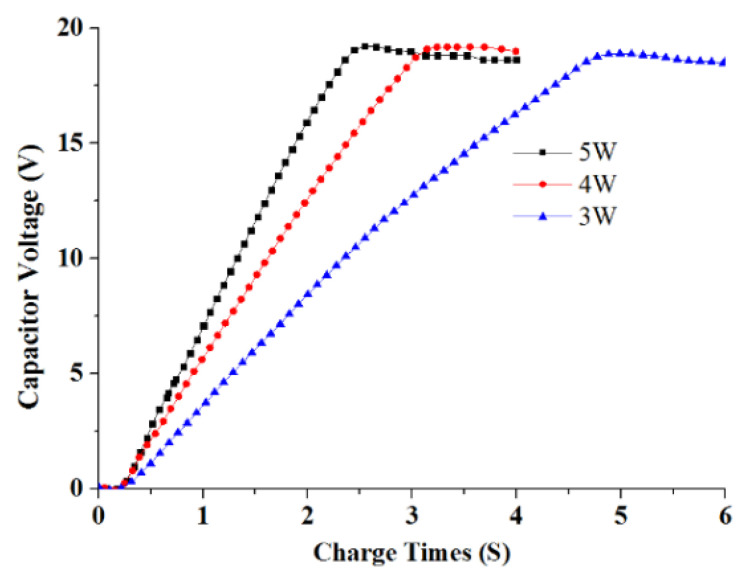
Curve of capacitor voltage and charging time.

**Figure 14 micromachines-13-00728-f014:**
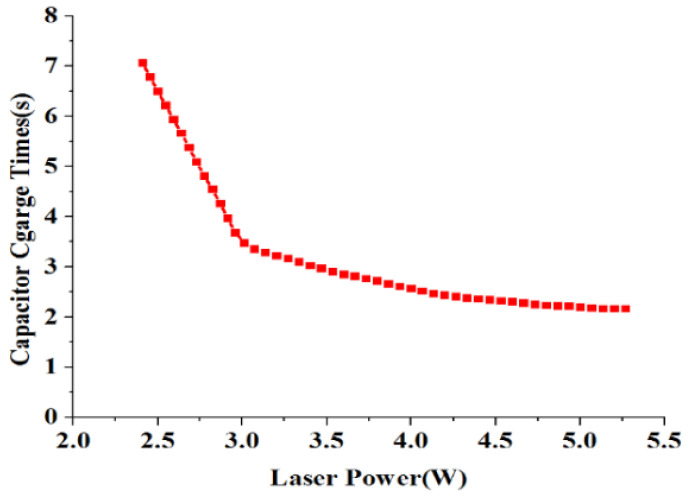
Curve of charging time and laser power.

**Figure 15 micromachines-13-00728-f015:**
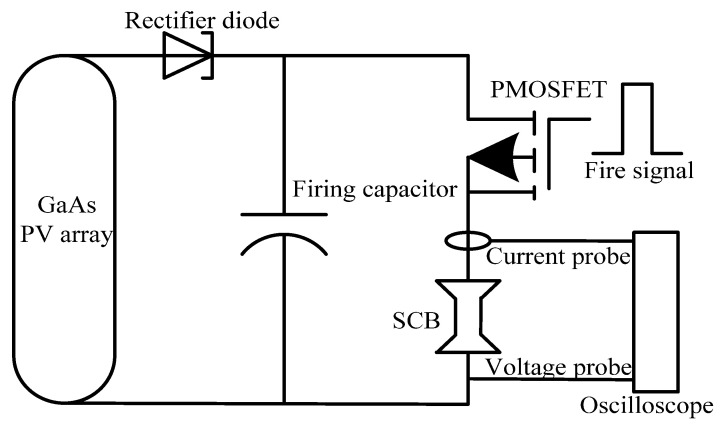
Schematic diagram of integrated energy device initiation performance experimental setup.

**Figure 16 micromachines-13-00728-f016:**
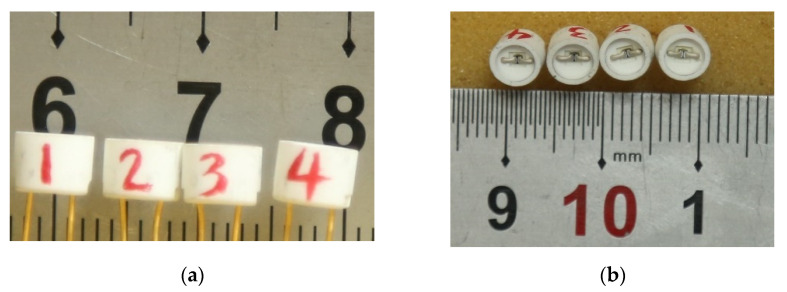
Structure of low-energy SCB. (**a**) Top view of the SCB; (**b**) side view of the SCB.

**Figure 17 micromachines-13-00728-f017:**
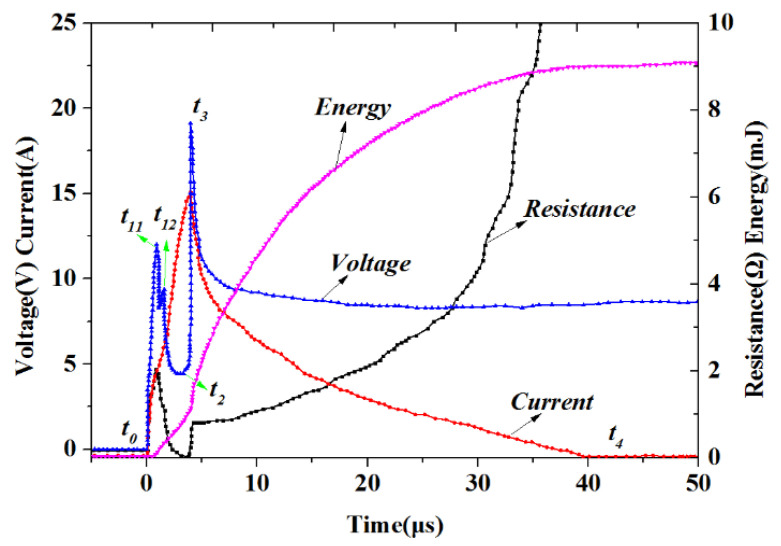
The initiation electric discharge characteristic curve.

**Figure 18 micromachines-13-00728-f018:**
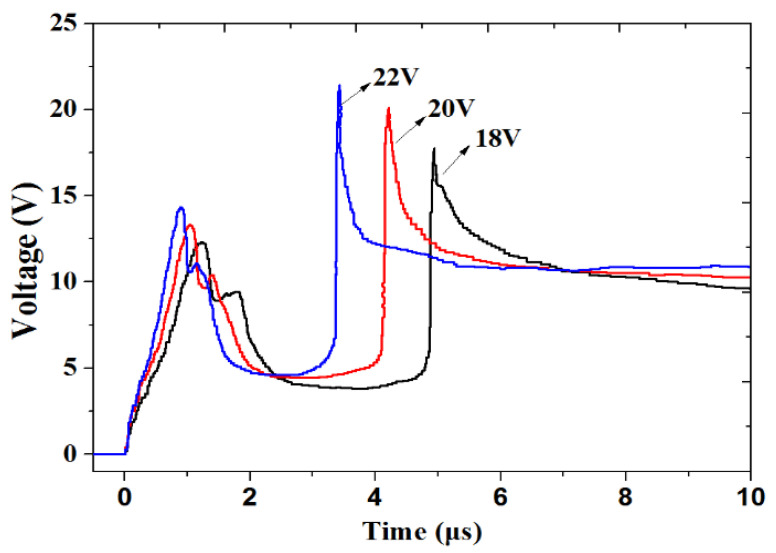
The initiation electric discharge characteristic curves with a different excitation voltage.

**Figure 19 micromachines-13-00728-f019:**
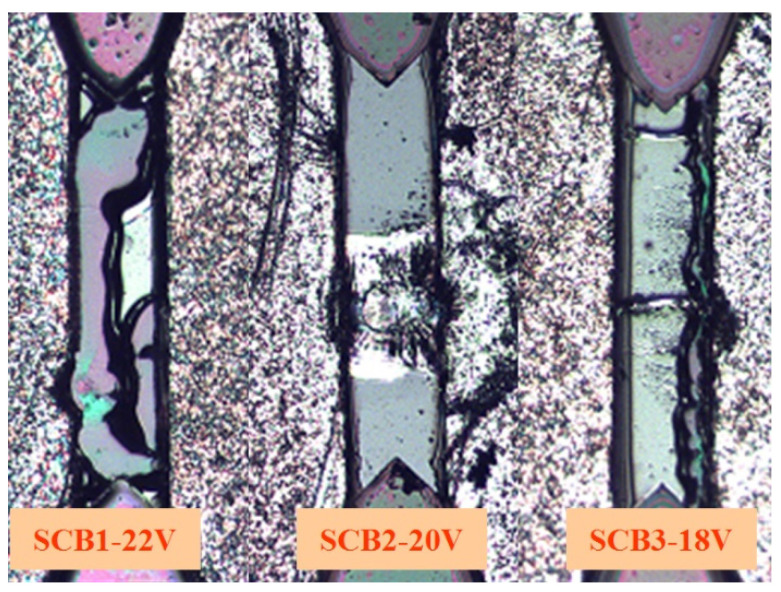
The result of LCEID with a different excitation voltage.

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
