# Peer review of "Design, Fabrication, and Characterization of a Laser-Controlled Explosion-Initiating Device with Integrated Safe-and-Arm, EMP-Resistant, and Fast-Acting Technology Based on Photovoltaic Power Converter"

_micromachines, 2022, doi:10.3390/mi13050728_

Round 1

Reviewer 1 Report

The manuscript is devoted to the first practical application of the phenomenon of photovoltaics in initiating devices. Therefore, the topic of the manuscript is very relevant.

The illustrations correctly reflect the data given in the manuscript.

  1. The reviewer recommends that the authors of the manuscript give the full names of energetic materials before their abbreviations on line 56 (HMX, THKP)
  2. The reviewer recommends that the authors of the manuscript provide examples of energetic materials used in LCEID in the initial charge, middle charge, and output charge (Fig. 2).
  3. The manuscript does not say anything about how reliably the semiconductor bridge initiates the initial charge. Therefore, authors are advised to correct this lack of text without fail.

Author Response

We are very grateful for the detailed feedback that you provide on our Micromachines submission (Manuscript ID 1641829 ). We think that the feedback resulted in valuable additions, changes, and improvements to our manuscript.

We have carefully addressed all the reviewer’s concerns. Please see below our replies. We hope he/she is satisfied with our answers. Changes highlighted in yellow have been made accordingly, in the revised manuscript and in the revised supplementary information.

Reviewer 2 Report

This study demonstrated the application of a GaAs photovoltaic laser power converter to the initiation system of rockets or other devices. Laser power converters have been previously investigated as the authors mentioned but applications in real life devices and systems are still ongoing. This manuscript has provided such performance demonstration combined with system optimization. So, in the reviewer's opinion, the manuscript is qualified to publish after revision.

Two major issues that require to be addressed are listed below.

1. A critical question is related to the reported open circuit voltage. High output voltages are required to power an electronic circuit such as a 15 V minimum mentioned in this work. Based on Figure 9 and the authors' analysis, the open circuit voltage (Voc) of one cell should be 10.75 V for this work. However, the theoretical open circuit voltage for a single GaAs cell is ~1.2 V mainly determined by the band gap. Indeed, the authors observed essentially high Voc. The irradiation power and change of band gap due to
doping may change the open circuit voltage but not likely on the scale the authors observed.

So, it is required to explain the fundamental reason for such high open circuit voltage and compare with literature reported values. In addition, it would be helpful to detail the measurement setup and conditions.

2. Figure 17 shows the firing process of the semiconductor bridge. The physical behaviors occurring after t3 are not fully clear. If the constant voltage is due to the capacitance behavior, why does the resistance increase rapidly? Please explain.

A few minor suggestions are listed below.

1, Line 17 and 23: Full terms should be given for these abbreviations such as EMP and ROM.
2, Line 70: The laser system, a critical component in the laser controlled explosion initiating device, should be detailed. Parameters include wavelength, power, spot size, and other info.

3, Line 197-204 (Figure 8 and Equation 1-4): The photovoltaic basics should be carefully referenced. All the symbols such as Id should be carefully annotated.
In addition, the dark current, series resistance, and shunt resistance, together with fill factor have not been analyzed and given in the manuscript. The reviewer wonders about the motivation of listing the equations of photovoltaic basics here. As such, the reviewer suggests the authors further provide these photovoltaic parameters of their GaAs cells in the discussion. Especially, Figure 11 shows a nearly 100% fill factor which is an excellent value that deserves further discussion.

4, Line 242 (Figure 11): Laser power and laser power density are different. 5.5 W here and in other places should be corrected to 5.5 W/cm2. The authors should double check the laser spot size, the size after beam expansion, and the effective irradiation size on the cells.
5, Line 275: What is the laser ablation limit for GaAs? A laser fluence value would be more useful than power density in here.

6, Line 304: Please provide the equations for calculating resistance and energy values.

Author Response

(The authors gave the same response as above.)
